# Validation of an Attributional and Distributive Justice Mediational Model on the Effects of Surface Acting on Emotional Exhaustion: An Experimental Study

**DOI:** 10.3390/ijerph18147505

**Published:** 2021-07-14

**Authors:** Alejandro García-Romero, David Martinez-Iñigo

**Affiliations:** 1International Doctorate School, Rey Juan Carlos University, 28922 Alcorcón, Spain; alejandro.garcia@urjc.es; 2Methodology in Behavioural Sciences Area, Rey Juan Carlos University, 28922 Alcorcón, Spain

**Keywords:** responsibility attribution, distributive justice, surface acting, emotional exhaustion

## Abstract

Previous research has shown that surface acting—displaying an emotion that is dissonant with inner feelings—negatively impacts employees’ well-being. However, most studies have neglected the meaning that employees develop around emotional demands requiring surface acting. This study examined how employees’ responsibility attributions of client behavior demanding surface acting influence employees’ emotional exhaustion, and the mediational role of distributive justice in this relationship. Relying on Fairness Theory, it was expected that employees’ responsibility attributions of client behavior demanding emotion regulation would be related to their perceptions of distributive injustice during the service encounter, which in turn would mediate the effects of responsibility attribution on emotional exhaustion. In addition, drawing on the conservation of resources model, we contended that leader support would moderate the impact of distributive injustice on emotional exhaustion. Two scenario-based experiments were conducted. Study 1 (*N* = 187) manipulated the attribution of responsibility for emotional demands. The findings showed that distributive injustice and emotional exhaustion were higher when responsibility for the surface acting demands was attributed to the client. A bootstrapping mediational analysis confirmed employees’ attributions have an indirect effect on emotional exhaustion through distributive justice. Study 2 (*N* = 227) manipulated responsibility attribution and leader support. The leader support moderation effect was confirmed.

## 1. Introduction

Numerous investigations have shown that the use of surface acting—expressing emotions to clients that do not match the employee’s feelings to meet the emotional demands of the work role [1]—has negative consequences for the employee’s work well-being [2]. While other regulatory strategies, such as deep acting or automatic regulation, are healthier for the employee [3], it is difficult to imagine a work context in which surface acting is not used. The different affective events in the work environments and the personal characteristics of individuals make it inevitable that, occasionally, the emotional experience of the employee will not align with emotional norms. The ubiquity of surface acting marks the study of the variables that mediate or moderate the relationship with work well-being as a relevant area in terms of employees’ psychological well-being.

Although the range of variables analyzed to explain the negative effects of surface on job well-being is wide [4,5,6,7], the role of situational factors has been neglected. Most studies have drawn on the idea that the imbalance between an employee’s available resources and the resources needed to meet the work role demands negatively impacts job well-being [8,9]. In the case of emotional demands at work, studies on the relationship between emotional regulation and psychological well-being have explained this relationship as being the result of the balance between the consumption of resources needed to regulate emotions, and the opportunities for recovering consumed resources [10]. Despite the widespread use of this explanation, reviews on the relationship between self-regulation of emotions and well-being have questioned it [11,12,13]. Among the situational factors that previous research has neglected, the meaning employees attach to client behaviors demanding surface acting, and the perception of distributive justice, has shown some empirical evidence in terms of their relevance to understanding under what conditions surface acting threatens employees´ well-being.

Recent studies have shown that the negative impact of surface acting partially depends on employees´ attributions for the emotional demands. Niven et al. (2019) [14] showed that the attributions employees make about the motives people hold to influence the feelings of others either positively or negatively are related to the quality of their relationship with leaders. Yagil (2020) [15] found that employees’ attributions related to the stability of client behavior (e.g., the employee’s perception that the client’s current behavior reflects the way he usually behaves) or the presence of egoistic motives can also contribute to increasing the employee’s perception of distributive injustice. However, to the authors’ knowledge, the role of responsibility attribution has not been explored, although responsibility attribution is especially relevant to employees’ responses to situational demands [16]. The attribution of responsibility has been identified as a necessary condition to perceive other people’s demands as unfair [17]. As a sine qua non condition to perceive distributive justice, an integrative model considering both variables can contribute to furthering our understanding of the impact of surface acting on well-being.

To illustrate the potential relevance of attribution of responsibility and distributive justice, consider two scenarios in which the employee of an air travel company interacts with an anxious passenger that has just missed their flight. Will the perception of distributive justice and the subsequent emotional exhaustion be the same when missing the flight is due to the employee forgetting to announce boarding, compared to when it is due to the passenger ignoring repeated boarding messages? 

When the interaction partner is considered responsible for the behavior demanding surface acting, employees´ perceptions of the effort required to regulate their emotions as distributively unfair will increase. On the contrary, when the employee is responsible for the client’s behavior (they forget to announce the flight), they can reinterpret their effort to regulate their own emotions as proportionate, considering the consequences of their own behavior (the passenger missing the flight). Consequently, the attribution of responsibility as being the client’s might be related to the employee’s assessment of distributive justice during the interaction, which in turn is related to the employee’s level of emotional exhaustion. 

The aim of our research is to validate a responsibility attribution and fairness model for the consequences of surface acting in terms of job well-being. The model integrates attribution of responsibility on surface acting demands, and its impact on distributive justice perception relationships, as predictors of employees’ emotional exhaustion (Figure 1). 

Drawing on fairness theory and its relationship with attributions, we propose that employees’ responsibility attributions for client behavior demanding surface acting will be related to their emotional exhaustion. Moreover, we expect that the impact of attributions on emotional exhaustion will be mediated by their relationship with the employees’ perceptions of distributive justice.

Two scenario-based experiments were conducted to test the model using structural equation modeling. The first manipulated the attribution of responsibility to test the mediation of distributive justice on its relationship with emotional exhaustion and the variables proposed in the model. The second study confirmed the robustness of the model by testing whether the presence of an external source of support (leader support) moderates the relationship between the perception of distributive injustice and job well-being, as predicted by the *conservation of resources model* [18,19]. According to this model, the availability of positive relationships (e.g., supervisor support) following an episode of surface acting can contribute to the recovery and conservation of the employee’s resources, ameliorating the negative impact on well-being [20,21], even if the employee’s attributions of the client behavior define the exchange as unfair. Considering that episodes of surface acting do not take place in a social vacuum, the second study considered the impact of other actors´ behaviors, not just the client, on employees’ perceptions of distributive justice following surface acting implementation. Previous research confirmed that restoration of distributive justice through economical compensation reduces the impact of surface acting on employees’ satisfaction [22]. The restorative effects of social support on the relationship between distributive justice and employees’ well-being following emotional regulation have not been tested.

### 1.1. An Attributional and Distributive Justice Model for the Relationship between Surface Acting and Emotional Exhaustion

#### 1.1.1. Responsibility Attributions and Emotional Exhaustion

Attributional models consider the role of an individual´s appraisal of negative events on their emotional experiences. In particular, [23] model of the Attributional Theory of Motivation and Emotion addresses the role of attributions in shaping the emotional experiences that derive from the demands of the environment [23,24]. Adopting a cognitive perspective on emotions, the model proposes that the emotional experience results from a temporal sequence of cognitions of increasing complexity. In the case of employee–client interactions, the presence of external demands triggers a primary appraisal that determines their relevance for the individual goals. When the employee’s inner feelings (i.e., anger towards the client) and display rules (i.e., service with a smile) are not aligned, it is primarily appraised as a demanding event that triggers an unspecific negative experience associated with emotional dissonance. Primary appraisal is a rather rapid and automatic response that is followed by secondary appraisals that often involve ego-related and more complex psychological mechanisms that are able to intensify or modulate the intensity of the emotional experience. Attributions of the demanding event are among those secondary appraisals used to understand the origin and consequences of the event, that modulate the individual’s experience. In organizational contexts, causality and control attributions contribute to explaining reactions to customer behaviors [14,25,26,27]. In the case of surface acting, the employee’s attribution of responsibility to the client for their behaviors demanding surface acting may intensify the employee’s experience of stress, and the subsequent emotional exhaustion. Based on this evidence, we expect:

**Hypothesis** **1** **(H1).**
*The mean level of emotional exhaustion will be higher when employees’ attribute the responsibility for client’s surface acting demanding behaviors to the client (hetero-attribution), when compared with employees who self-attribute responsibility for the client’s behaviors demanding surface acting (self-attribution).*


#### 1.1.2. Responsibility Attribution and Distributive Justice

Drawing on *fairness theory* [17], the effects of attribution on well-being may be explained through the relationship between responsibility attributions and distributive justice as secondary appraisal processes. According to this theory, “the process of accountability, or how another social entity comes to be considered blameworthy, is fundamental to justice” [17] (p. 3).

Within the study of distributive justice, the responsibility attribution towards the interaction partner’s behavior is one of the main antecedents of the assessment of what the person does, on the balance of the contributions and outcomes from the social exchange during the interaction [17,28,29,30,31]. Ref. [17] showed that when behaviors were perceived as avoidable and that they should have been avoided, but were not despite the existence of better alternatives, perceptions of injustice intensified. In addition, contingency models of distributive justice posit that appraisal of the claimant’s needs is central to the perception of fairness during social encounters. When claims arise from internally controllable causes, allocators devalue the claimant’s deservingness and withhold resources from the social exchange [32]. The empirical evidence supports that when attributions of responsibility are made on the other person, the exchange is perceived as more unfair than when the responsibility falls on oneself [28,29,33,34]. 

Something similar occurs when clients are responsible for behaviors demanding employees’ surface acting. When employees perceive that the client is responsible for the behavior demanding surface acting, they may perceive that regulation of their emotions is unwarranted. Under these circumstances, withdrawing resources or increasing client costs (retribution) are natural reactions to restore equity in social exchange interactions [16,35]. Nevertheless, the employee’s ability to withhold resources or retaliation during service delivery interactions is constrained by the organization’s emotional norms, requiring a positive interaction even when the client “does not deserve” this attention (i.e., “the customer is always right”). As mechanisms for distributive justice restoration are blocked by organizational display rules, a negative impact on distributive justice can be expected when employees attribute to the client the responsibility for behaviors that demand surface acting, without being able to withhold resources or retaliate.

On the other hand, when the interaction partner cannot be held accountable for the behavior that demands surface acting, the perception of unfairness decreases. If responsible for the client’s behavior, employees will underestimate their contribution to the social exchange. For instance, the perception of clients as not responsible for the surface acting demand decreases the employees’ assessment of the cost of unauthenticity [15]. From a fairness theory perspective, the employee’s self-attribution of responsibility for the client’s behavior will increase their estimation of the costs for the clients, and the employee’s willingness to enact a restorative action. The effort to regulate their emotion will be perceived by employees as a process of restoring fair exchange conditions, thus reducing the negative impact on their emotional exhaustion.

In this scenario, we hypothesize:

**Hypothesis** **2** **(H2).**
*The level of perceived distributive justice will be higher when employees self-attribute the responsibility for client’s surface acting demanding behaviors, when compared with interactions where the responsibility for those behaviors is attributed to the client (hetero-attribution).*


#### 1.1.3. Distributive Justice and Emotional Exhaustion

The perception of distributive injustice involves an estimation of imbalance between consumed and recovered resources [35]. This imbalance entails the judgment that the amount of resources consumed during an interaction is higher than the recovered input. Experimental evidence suggests that individuals’ beliefs and expectations regarding the amount of self-regulation resource left by the regulation of emotions contribute to explaining their level of emotional exhaustion [36,37,38]. When the expectation that resources are abundant after the regulation of emotion is primed, the level of exhaustion remains stable, contrary to when the idea that resources are scarce is primed [38].

The perception of a negative exchange balance may generate the expectation that available resources for future interactions will be scarce, increasing emotional exhaustion. Studies conducted on the relationship between equity and work well-being show that when the social exchange is perceived as distributively unfair, psychological well-being decreases [21,39,40]. In the case of surface acting, Ref. [41] found that perceptions of distributive injustice following surface acting increase employees’ emotional exhaustion. Based on these results, we expect:

**Hypothesis** **3** **(H3).**
*The perceptions of distributive injustice will be negatively correlated with the expected emotional exhaustion following unfair encounters.*


### 1.2. Responsibility Attribution, Distributive Justice, and Emotional Exhaustion

Considering that organizational norms regarding emotional regulation apply to all employee–client interactions (e.g., service with a smile rule), the attributions that employees make about the behavior of clients that demand surface acting might explain the perception of greater or lower justice in the social exchange with the client, which in turn relates to their level of emotional exhaustion.

When employees attribute responsibility to the client (*hetero-attributions*), they will find it unfair to apply a rule that forces them to regulate their emotions, which in turn increases their emotional exhaustion. On the contrary, it is possible that, if the employees self-attribute the responsibility (*self-attribution*), they will consider it fairer that they must regulate their emotions to serve the client, reducing the impact on employees’ emotional exhaustion. We hypothesize:

**Hypothesis** **4** **(H4).**
*The negative effects of responsibility on emotional exhaustion will be mediated by the employees’ perceptions of distributive justice.*


### 1.3. Distributive Justice, Leader Support, and Emotional Exhaustion

As mentioned above, the *Conservation of Resources model* has been widely applied to explain the negative impact of surface acting on job well-being. According to this model, individuals attempt to obtain, sustain, and preserve their resources, including self-regulation resources, to prevent exhaustion [18]. Positive interpersonal relationships are one of the main sources of resource recovery [20,42]. Although following surface acting the likelihood of resource recovery from the client´s response is low because inauthenticity in the emotional display reduces the likelihood of positive responses [43], previous research has shown that reactions from other actors observing a particular social exchange contribute to recovering the resource consumed during the interaction with the client. For example, Ref. [44] posited that when employees perceive that service encounters are distributively unfair, they turn to the organization for recovering the resource consumed during interactions with clients. For surface acting, Ref. [22] found that financial rewards associated with emotion regulation buffer the negative consequences of emotional demands on employees’ job satisfaction. It could be the case that socioemotional resources (e.g., social support) have the same buffering effects on the relationship between unfairness and emotional exhaustion [45]. We expect:

**Hypothesis** **5** **(H5).**
*Support from a leader will moderate (reduce) the impact of distributive injustice on the expected emotional exhaustion following surface acting episodes.*


## 2. Study 1

### 2.1. Materials and Methods

#### 2.1.1. Participants

The sample consisted of 187 participants. We used a snowball technique, requesting students from a university campus to disseminate the scenarios and questionnaires of the study through their social networks. A total of 187 employed participants completed the online questionnaire. The mean age was 27.93 (SD = 10.47), and 79.7% of the participants were female and 20.3% male. The percentage of participants with experience facing the public was 81.1%, compared to 18.9% with no experience. The corresponding ethics committee approved the project, and informed consent was requested from the participants.

#### 2.1.2. Procedure

As part of the study, the participants had to read two scenarios describing the interaction between an employee of an airline and a passenger. In both scenarios the passenger requested to board the plane after the doors had been closed. During the interaction, the employee’s behavior was described as kind, despite experiencing negative emotions. Attribution of responsibility was manipulated by changing the reason for missing the flight. In the employee’s responsibility scenario (self-attribution), the passenger misses the flight because the employee forgets to announce a change in the departing time of the flight. In the client’s responsibility scenario (hetero-attribution), the passenger missed the flight because they got held up at a wine tasting in the duty-free area. The scenario describes how the employee shows a positive emotional expression, despite of the negative feelings deriving from the client´s demanding behaviors (*the employee kindly explains how to proceed. However, he has to make an effort to be nice and hide the negative feelings from him, and the stress caused by the situation*).

The participants were required to take the employee’s role while reading the scenarios and making their assessment. The order of presentation of both scenarios was counterbalanced. After the completion of each scenario, participants filled in an online questionnaire with the study measures.

#### 2.1.3. Instruments

Sociodemographic variables: participants were asked about age and gender.

*Expected Emotional Exhaustion.* This was measured using eight items from the Spanish version [46] of the Maslach Burnout Inventory Manual [47]. Participants were instructed to rate on a 5-point scale (1: *Nothing*–5: *A lot*) how emotionally exhausted the employee would feel after interacting with passengers such as the one described in the scenario during a working day (sample item: *To what extent will the employee feel emotionally drained?*). The internal consistency of this scale was α = 0.94.

Distributive Justice. This was measured using the six items from Spanish version [48] of the Organization Justice Scale [49] (sample item: To what extent did the passenger’s behavior during the interaction reciprocate the employee’s effort to be kind and understanding?), and one ad hoc item asking the participants to compare the employee’s and passenger’s input during the interaction using a 5-point scale (comparing the employee’s and client’s effort to regulate his emotions. 1: The employee’s effort was much greater that the passenger’s effort; 5: The passenger’s effort was much greater that the employee’s effort). For the rest of the items, the participants were asked to rate to what extent they agreed with each proposition on a 5-point scale, ranging from 1 (Nothing) to 5 (Totally). The internal consistency of this scale was α = 0.89.

*Control variables.* An individual’s previous experience of service delivery has been shown to be related to surface acting’s consequences [50], thus previous experience was controlled for. Participants were asked to indicate if they had previously worked in the service sector. Because *psychological effort* is related to the amount of self-regulation resource consumed during emotion regulation, which in turns affects emotional exhaustion [51]; their effects were controlled for in the model test. Participants were asked to rate on a 5-point scale, ranging from 1 (*Nothing*) to 5 (*A lot*), the employee’s effort to regulate their emotions. We used seven items from the Emotional Effort Scale [52] (sample item: *The employee has put great effort into controlling his emotional expression?*). The internal consistency of this scale was α = 0.80.

#### 2.1.4. Experimental Controls

*Responsibility Attributions.* To check the experimental manipulation effectiveness, the attribution of responsibility for the client’s behaviors demanding emotion regulation was measured for each scenario. We adapted the Occupational Attributional Style Questionnaire [53]. The scale includes 11 items (sample item: *To what extent was the situation motivating the passenger claim provoked by himself?*). Participants were asked to rate on a 5-point scale (1 *Not at all* to 5 *Totally*) to what extent they agreed with each item’s proposition. Lower scores indicated client responsibility and higher scores indicated employee responsibility. The internal consistency of this scale was α = 0.92.

### 2.2. Analyses

First, to confirm the effectiveness of the experimental manipulation, a *t*-test for related samples was conducted comparing the mean levels of responsibility attribution for both scenarios.

The hypothesis for the differences between experimental conditions on perceived distributive justice and emotional exhaustion were also tested with *t*-tests for related samples.

Structural equation modeling (SEM) was conducted to test whether distributive justice mediated the relationship between responsibility attributions and expected emotional exhaustion [54]. To avoid dependency in the data, we tested a cross-sectional model based on the response of all participants to the scenario (self- vs. hetero-attribution) they read first. Additional evidence on the validity of the model was obtained by conducting the same analysis with the participants’ scores for the scenario they read second. Distributive justice was regressed on attribution of responsibility and emotional exhaustion to test the hypothesized indirect effects [55]. Confidence intervals around the point estimation of indirect effects were computed using bootstrapping resampling with 5000 samples. Indirect effects were considered significant when zero was not included in the 95% confidence interval. The effects of psychological effort and previous experiences with clients on emotional exhaustion were controlled for.

### 2.3. Results

Descriptive analyses for the variables in the study are summarized in Table 1.

*T*-test for dependent sample on responsibility attribution over client’s behavior demanding emotional regulation confirms the experimental manipulation was effective. The mean level of responsibility attribution on the client for self-responsibility scenario (M = 3.55, Sd = 0.63) was significantly higher, *t* (186) = 29.21, *p* < 0.001, when compared with the mean level of the hetero-responsibility scenario (M = 1.81, Sd = 43), d = 5.69.

The difference in emotional exhaustion between participants on the self-responsibility (M = 3.39, Sd = 0.97) and the hetero-responsibility (M = 3.82, Sd = 0.81) scenarios was significant, *t*(186) = −6.57, *p* < 0.001, d = 0.48. Emotional exhaustion was higher when participants reported hetero-attributed responsibility compared with their scores when responsibility was self-attributed. Hypothesis 1 was thus supported.

Results from the *t*-test for dependent samples supported that the mean level of distributive justice was significantly lower, *t*(186) = 22.13, *p* < 0.001, d = 1.98, when responsibility was hetero-attributed (M = 1.91, Sd = 0.67), compared to when responsibility was self-attributed (M = 3.31, Sd = 0.74). Hypothesis 2 was thus supported.

Bivariate correlations confirmed the expected relationship between responsibility attribution, distributive justice, and expected emotional exhaustion, except for the relationship between attribution of responsibility and distributive justice when the scenarios were first presented (see Table 2).

Before testing the model, the CFA of the measurement model for the variables included in the study was conducted. The three-factor model, including attribution of responsibility, distributive justice, and emotional exhaustion for scores from the scenario presented first showed excellent goodness of fit (χ^2^ = 289.33, df = 259, *p* = 0.10; RMSEA = 0.03; SRMR = 0.06; CFI = 0.99; TLI = 0.99) [56]. Fitness indices for scores from the scenario presented second were good (χ^2^ = 397.82, df = 246, *p* = 0.001; RMSEA = 0.06; SRMR = 0.06; CFI = 0.97; TLI = 0.96).

To test the mediational model, we separately conducted analyses for all participants’ scores when the scenarios were presented first and for the same scores when the scenarios were presented second.

The results confirmed a significant positive direct effect from attribution of responsibility to distributive justice when the scenarios were presented first (b = 0.69, SE = 0.06, *p* < 0.001) and second (b = 0.75, SE = 0.04, *p* < 0.001) (see Table 3). For the direct effects of distributive justice on expected emotional exhaustion, results from both conditions confirmed a significant negative effect (first place b = −0.12, SE = 0.06, *p* = 0.03, second place b = −0.19, SE = 0.05, *p* < 0.001) (see Table 3). H3 was thus supported.

For the indirect effects of responsibility attribution on the expected emotional exhaustion, the results showed a significant indirect effect through distributive justice for the scores when the scenarios were presented first (b = −0.11 *p*= 0.03; Bootstrap = 5000 CI 95% [−0.23, −0.01)]; k^2^ = 0.09) and for the scores from the scenarios appearing second (b = −0.14 *p* < 0.03; Bootstrap = 5000 CI 95% [−0.26, −0.01]; k^2^ = 0.11) (see Table 3). Hypothesis 4 was thus supported.

## 3. Study 2

The second study was designed to replicate the evidence for the proposed model and to test the hypothesis of the meditational role of leader support on the relationship between distributive justice and emotional exhaustion.

### 3.1. Materials and Methods

#### 3.1.1. Participants

The sample consisted of undergraduate students who volunteered to participate in a scenario-based experiment. A total of 227 participants completed the online questionnaire. The mean age was 22.22 (SD = 8.0), and 76.2% of the participants were female and 23.2% male. The percentage of participants with experience facing the public was 67.8%, compared to 32.2% with no experience. The corresponding ethics committee approved the project, and informed consent was requested from the participants.

#### 3.1.2. Experimental Design

A 2 × 2 mixed design was used, with responsibility attribution as the intrasubject factor (self vs. hetero) and leader support as the interpersonal factor (support vs. no support).

#### 3.1.3. Procedure

Participants were randomly assigned to the leader support conditions. For each condition in this factor, they read the self and hetero responsibility scenarios described in Study 1. The order of presentation of both scenarios was counterbalanced. Leader support was manipulated by adding a sentence describing the reaction from the leader after the interaction with the passenger. In the support condition, the leader observes the interaction and when the passenger leaves the scene, approaches the employee showing empathy and congratulating them on their management of the situation. For the nonsupport situation, the leader observes the interaction but once it is finished remains engaged with other tasks, without interacting with the employee. After the completion of each scenario, participants filled in an online questionnaire with the study measures.

#### 3.1.4. Instruments

In addition to the measures from Study 1, leader support was measured using 10 items adapted from the Survey of Perceived Organizational Support [57]. The items reflected how the employee would perceive the degree of leader support after interacting with the passengers, considering the behavior of the supervisor described in the different scenarios (sample item: *The leader really cares about employee’s well-being?*). Participants were instructed to rate to what extent they agreed with each proposition on a 5-point scale ranging from 1 (*Nothing*) to 5 (*Totally*). The internal consistency of this scale was α = 0.88.

Internal consistency for all the scales in the study ranged from 0.78 to 0.96.

### 3.2. Analyses

*T*-tests for related (responsibility attribution) and independent (leader support) samples were conducted to confirm the effectiveness of the experimental manipulation. A mixed 2 × 2 ANOVA was conducted to confirm the main effects of responsibility attribution on distributive justice and emotional exhaustion and assess the possible interaction effect between responsibility attribution and emotional exhaustion.

As for Study 1, structural equation modeling was separately conducted for the scores when the scenarios were presented in first and second place, to test the indirect effects of responsibility attribution on expected emotional exhaustion as mediated by distributive justice. Additionally, to test the leader support moderation on the relationship between distributive justice and emotional exhaustion, leader support and its interaction term with distributive justice were included as predictors of emotional exhaustion [54].

### 3.3. Results

Table 4 shows the descriptives forthe variables in study 2.

The *t*-test for dependent samples on responsibility attribution (self M = 3.6, Sd = 0.64; hetero M = 1.89, Sd = 0.63) confirmed the experimental manipulation was effective, *t* (226) = 26.89, *p* < 0.001, d = 2.57. The *t*-test for independent samples confirmed the experimental manipulation of leader support (support M = 3.30, Sd = 0.54; No support M = 2.02, Sd = 0.61) was effective, *t* (225) = −16.93, *p* < 0.001, d = 0.26).

The results from mixed ANOVA showed the mean level of distributive justice was significantly higher, F(1, 225) = 353.33, *p* < 0.001, for the self-attribution condition (M = 3.0, Sd = 0.83), when compared with the hetero-attribution condition (M = 1.9, Sd = 0.63). Hypothesis 1 was supported. The interaction effect between leader support and responsibility attribution was not significant, F(1, 225) = 0.001, *p* = 0.95.

As expected, the mean level of emotional exhaustion was significantly higher, F(1, 225) = −12.32, *p* < 0.01), for the hetero-attribution condition (M = 3.47 Sd = 0.89) when compared with the participants in the self-attribution condition (M = 3.27 Sd = 0.87). Hypothesis 2 was supported. The interaction effect between leader support and responsibility attribution was not significant F(1, 225) = 0.06, *p* = 0.81.

Bivariate correlations were computed separately for the distributive justice and emotional exhaustion relationship scores when the two scenarios were presented in first place and when they were presented in second place. The bivariate correlation was significant only when the scenarios were presented in second place (r = −0.32, *p* < 0.001). 

As the bivariate correlation between distributive justice and emotional exhaustion was only significant for the second scenario presented, the model was tested only for those scores. The results confirmed a significant positive direct effect from attribution of responsibility to distributive justice (b = 0.76, SE = 0.05, *p* < 0.001).

For the direct effects of distributive justice on expected emotional exhaustion, the results confirmed a significant negative effect (b = −0.65, SE = 0.08, *p* < 0.01). Hypothesis 3 was supported. The responsibility attribution indirect effects were significant (b = −0.50 *p* < 0.001; Bootstrap = 5000 CI 95% [–0.77, −0.25]; k^2^ = 0.46). Hypothesis 4 was thus supported. The coefficient for the distributive justice and leaders’ support interaction term was also significant (b = 0.18, SE = 0.08, *p* < 0.001; see Table 5). 

To test whether the moderation effects confirmed that the positive association between distributive injustice and expected emotional exhaustion can be attenuated when leader support is increased, we estimated the regression coefficients for different levels of leader support. The results showed that the conditional effects of attribution responsibility decreased when leader support levels increased and were significant only at a lower level of support (see Table 6). Hypothesis 5 was thus supported.

## 4. Discussion

The main goal of our study was to analyze the unexplored effects of employees’ responsibility attributions for client behavior demanding surface acting, which are related to employee emotional exhaustion, and the mediational role of distributive justice in this relationship. As predicted by Fairness Theory [17], our results showed that employees’ self-attribution of responsibility for the client’s behavior is positively related to employees’ perception of distributive justice, which in turn is negatively related to employees’ emotional exhaustion. The study supports the mediational role of distributive justice on the relationship between responsibility attributions and emotional exhaustion. Additionally, the results confirm that the indirect effects of responsibility attributions through distributive justice are moderated by the presence of leader support.

The mediation effects of distributive justice on the attribution of responsibility and emotional exhaustion relationship were significant, with a large effect size. In Study 2, the indirect effects were significant only for the scenarios presented second. The inconsistency in the results could be explained as an effect of the order of scenarios. One of the explanations for the distributive justice effects on emotional exhaustion is based on the idea that people conform to expectations around the amount of resources available for future interactions, based on results from previous encounters. These effects rely on the temporal dimension, and they appear in future interactions. When participants dealt with the scenario presented first, they did not have a “history” of previous interactions and lacked an element of comparison, so the attribution of responsibility and the level of distributive justice may not have affected their level of emotional exhaustion after the first interaction, as it did for their estimation of emotional exhaustion for the scenario they read second. For the second scenario, they were able to compare their actual results with the outcomes from the encounter described in the previous scenario. Future research is needed to test this explanation.

Despite these limitations, we believe that the present study provides meaningful advances in the understanding of the relationship between surface acting and employees’ well-being. Both its conceptual and empirical approaches complement earlier research and participate, through their integrative perspective, in the convergence of different perspectives (attribution theory, fairness theories, and the conservation of resources model) on the determinants of employees’ job well-being.

By identifying one of the determinants of perceived distributive justice, responsibility attribution, our study supplements research that shows how employees’ perceptions of distributive justice in their interactions with clients explains the effects of surface acting on emotional exhaustion [22,41,58,59].

This study also supported the contribution of responsibility attribution to explain the indirect effects of surface acting on emotional exhaustion, besides and above the impact of the amount of self-regulation resources spent to meet emotional demands. These findings complement the recently questioned predictions from the strength of the self-regulation model [10]. Previous studies found that the attribution of responsibility to the client increases the level of emotional dissonance in the employees’ experience and, consequently, the amount of effort demanded to suppress the expression of negative emotions [34,51]. The possibility of responsibility affecting emotional exhaustion because of the increase in the amount of psychological effort was ruled out by controlling its effects on emotional exhaustion. This study supports the idea that, besides the amount of effort employees put into the suppression of their emotions, as proposed by the self-strength regulation model, the interpretation of the causes is relevant to explain emotional exhaustion.

### Limitations and Future Directions

Although the use of fictional scenarios is a widely used approach to the study of attributions and distributive justice [60,61,62], future research should replicate our results in natural settings or with more realistic simulations (e.g., interaction with confederates). Participants’ responses to scenarios could be influenced by social desirability bias, reflecting the social prescription of social interactions rather than the actual experience and consequences of social exchanges during service delivery encounters.

The use of self-reported measures could be partially responsible for the size of the relationships between some of the variables, because of common-method variables. The incorporation of behavioral measures would reduce the risk of common-methods bias. Emotional exhaustion could be indirectly measured by different behavioral indices (e.g., persistence with an unsolvable task) and psychological effort estimated from psychophysiological responses (e.g., heart rate variability). Although CFA does not preclude common method variance, the goodness-of-fit traces for the measures included in the study and the nonsignificance of some of the expected relationships suggest its impact on the results is acceptable.

This study focused exclusively on the distributive dimension of justice. Previous research has already established the relevance of other dimensions, such as interpersonal justice [30]. Although distributive justice has been shown to be the most important dimension for the analysis of job well-being, joint analysis of the different dimensions of justice (procedural, interpersonal, and distributive) would contribute to a better understanding of the impact of surface acting on employees’ well-being.

Other dimensions may contribute to further understanding how responsibility attributions for behaviors that demand emotion regulation affects emotional exhaustion, as suggested by a multifocal perspective on organizational justice. Future research should analyze whether attributions of responsibility are related to perceptions of interactional justice. When employees blame a client for the situation that provokes the emotional demand, the client’s behavior could increase the employee perception of interactional injustice, compared to an encounter where the responsibility for the same kind of behavior is attributed to the employee. It is also possible that the perception of justice in the procedures to manage unwarranted demands from the client can bring some light into the attributions of responsibility and the emotional exhaustion relationship. When organizational norms to deal with clients are applied to interactions where the employees perceive the demands as unwarranted, they can assess these procedures as unfair and feel themselves to be in a vulnerable position that ignores their interests, or the difficulties they deal with, during these encounters. Future research is needed to test these hypotheses.

Finally, other attributional dimensions need to be considered. Recent studies on surface acting [15] and interpersonal regulation of emotions [14] have shown that the attributions that people make about motives to influence the feelings of others positively or negatively are related to the quality of the leader–member relationship. Employees’ attributions regarding the stability of client behavior (e.g., the employee’s perception of the client’s current behavior reflects the way he usually behaves) or the presence of egoistic motives (e.g., the employee believes the passenger had no intention to take the flight and is pretending he missed the flight to claim compensation) can also contribute to increasing the employee’s perception of distributive injustice.

A joint analysis of employees’ attributions regarding the origin of emotional demands requiring surface acting and the impact on the evaluation of distributive justice could provide a more complete understanding of the processes through which the suppression of emotions has a negative impact on well-being at work, and when it will occur. Understanding the role of another source of positive feedback also offers additional evidence of the mechanism involved in the relationship between attribution, distributive justice, and emotional exhaustion.

## 5. Conclusions

Overall, our study suggests that a better understanding of the relationship between surface acting and emotional exhaustion could be reached if employee–client encounters are framed in the broader context of social exchange relationships at work. Our results are compatible with the notion that the extent to which employees perceive they are responsible for the emotional demand, and not only the effort required to regulate their emotions, influences their emotional exhaustion through its effects on their perception of distributive justice. This idea was already contemplated in Hochschild’s original formulation of the concept of emotional labor [1], and its consequences for the well-being of employees. The imposition of organizational norms on emotional regulation includes restrictions on the usual processes that modulate social exchange. In particular, the need to satisfy the norms of expression restricts the possibility of applying strategies for restoring justice in the exchange, such as retribution [35], when the emotional demand is attributed to the client [16]. Expanding the control of organizations over the emotional dimensions of behavior (“taylorization of emotions”) has been proposed as the primary explanation for the deleterious effects of emotional regulation on job well-being [63]. The results of both studies suggest the utility of recovering this perspective.

This study also supports the role of resource recovery through positive relationships with other actors, as proposed by the *conservation of resources model* [18]. Leader feedback ameliorates the negative impact of surface acting, supporting the idea of resource restoration through positive interactions with observers of the employee–client interactions.

Besides its theoretical relevance, some practical conclusions can be made based on this research. An organization’s emotional norms should be reconsidered in light of our results. Requesting that employees regulate their emotions when a client makes unwarranted demands endangers their well-being, even when impolite or abusive behaviors are absent. Since “service with a smile” is a well-established and barely questioned norm for many organizations trying to maximize clients’ positive experiences during service encounters, alternative actions should be required to buffer employees’ well-being. Our results suggest organizational support through leaders, when employees must deal with a client’s unwarranted demands, could ameliorate the negative impact on employees’ well-being. Explicit acknowledgement of these situations could significantly reduce the impact of surface acting when compensation from the client is not available [21,22].

## Figures and Tables

**Figure 1 ijerph-18-07505-f001:**
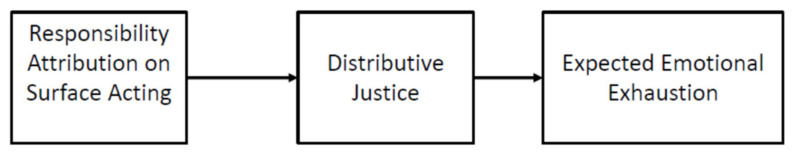
Responsibility attributions and distributive justice model for the surface acting and emotional exhaustion relationship.

**Table 1 ijerph-18-07505-t001:** Means and standard deviations for the study variables in each condition.

	Self-Attribution		Hetero-Attribution	
Variable	Mean	Sd	Mean	Sd
Effort	3.03	0.80	2.96	0.73
Attribution	3.55	0.63	1.81	0.43
Distributive Justice	3.31	0.74	1.91	0.67
Expected EE	3.39	0.97	3.82	0.81

Expected EE = expected emotional exhaustion.

**Table 2 ijerph-18-07505-t002:** Bivariate correlations between the study variables for presentation of the first and second scenario.

	1	2	3	4	5	6	7	8	9
**First Scenario**									
1. Experience									
2. Effort	0.03								
3. Attribution	0.19 *	0.17 **							
4. Distributive Justice	0.09	−0.07	0.66 *						
5. Emotional Exhaustion	−0.03	0.43 *	−0.07	−0.17 **					
**Second Scenario**									
6. Experience									
7. Effort						0.10			
8. Attribution						−0.16 **			
9. Distributive Justice						−0.08	−0.09	0.78 *	
10. Expected EE.						0.12 *	−0.52 *	−0.37 *	−0.36 *

* *p*< 0.01; ** *p*< 0.05. Expected EE = expected emotional exhaustion.

**Table 3 ijerph-18-07505-t003:** SME model of distributive justice as a mediator between attributions of responsibility and expected emotional exhaustion (N = 187).

	First Scenario		Second Scenario	
Variables	Distributive Justice	EE	Distributive Justice	EE
Tenure	0.12 (0.12)	−0.06 (0.12)	0.10 (0.12)	0.07 (0.14)
Effort	−0.05 (0.07)	0.48 (0.07)	0.02 (0.06)	0.63 (0.07) **
Attribution	0.69 (0.06) **	−0.02 (0.08)	0.75(0.04) **	−0.15 (0.08)
Distributive Justice		−0.16 (0.08) *		−0.19 (0.09) *
R^2^ total	0.45	0.24	0.62	0.38
Indirect Effect		−0.11 CI 95%		−0.14 CI 95%
	Bootstrap = 5000 [−0.23, −0.01]		Bootstrap = 5000 [−0.26, −0.01]	

* *p* < 0.05, ** *p* < 0.01; EE = expected emotional exhaustion.

**Table 4 ijerph-18-07505-t004:** Means and standard deviations for the study variables in each condition.

	Self-Attribution				Hetero-Attribution			
	No Support		Support		No Support		Support	
	M	Sd	M	Sd	M	Sd	M	Sd
Variable								
Effort	3.19	0.69	2.99	0.82	3.13	0.73	3.25	0.90
Attribution	2.24	0.36	2.26	0.39	3.6	0.62	3.59	0.65
Distributive Justice	2.94	0.88	3.05	0.78	1.83	0.67	1.94	0.59
Expected EE	3.57	0.88	3.01	0.76	3.79	0.89	3.19	0.79

**Table 5 ijerph-18-07505-t005:** SME model of social support as moderator of indirect effect of responsibility on expected emotional exhaustion with distributive justice as mediator (*N* = 227).

	Second Scenario	
Variables		
Tenure	0.10 (0.12)	
Effort	0.14 (0.07)	
Attribution	0.11(0.08) **	
Distributive Justice	−0.70 (0.08) *	
Social Support	−0.38 (0.05) *	
SSxDJ	0.17 (0.02) *	
R^2^ total		0.56
Indirect Effect		−0.34 CI 95%
	Bootstrap = 5000 [−0.69, −0.24]	

* *p* < 0.05, ** *p* < 0.01; SSxDJ = Social Support × Distributive Justice.

**Table 6 ijerph-18-07505-t006:** Conditional effect of responsibility attribution on expected emotional exhaustion at different values of leader support.

Level of Leader Support	Effect	SE	*t*	*p*	Bootstrap = 5000
1.50	−0.38	0.09	−3.96	0.001	[−0.56, −0.19]
2.70	−0.15	0.08	−1.88	0.06	[−0.31, 0.01]
3.60	0.01	0.11	0.12	0.91	[−0.20, 0.22]

## Data Availability

The data that support the findings of this study are available on request from the corresponding author, D.M.I.

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
