# Peer review of "Validation of an Attributional and Distributive Justice Mediational Model on the Effects of Surface Acting on Emotional Exhaustion: An Experimental Study"

_ijerph, 2021, doi:10.3390/ijerph18147505_

Round 1

Reviewer 1 Report

See the comments in attached file. 

Author Response

Reviewer´s comment: The main concern of this study fact that author(s) did not provide us proper research gap of this study. Previous studies are not enough for this manuscript.

Please add more previous studies in the introduction part.

Authors´ answer: The introduction has been rewritten and all the previous studies found in the review on the scientific literature referenced. The research gap (previous neglection of the attributional dimension and the distributive justice) and the paper contribution have been highlighted.

Reviewer´s comment: Author(s) used the COR model, I believe this is not a model this is a conservations of resources theory (Hobfoll, 1989). Author(s) need to write the proper way. I did not understand about the proper justifications of conservations of resources theory. Please specified the proper way of COR theory.

Authors´ answer: The inclusion of the COR in the paper has been rewritten and included in section 1.3. Also, the suggested reference has been added.

Reviewer´s comment: Author(s) needs to follow and must cite this paper and see how you can COR theory in emotional exhaustion.

References: Akram, Z., Li, Y., & Akram, U. (2019). When employees are emotionally exhausted due to abusive supervision. A conservation-of-resources perspective. International journal of environmental research and public health, 16(18), 3300.

Authors´ answer: The inclusion of the COR in the paper has been rewritten and included in section 1.3. Also, the suggested reference has been added.

Reviewer´s comment:  In the introduction part, author(s) do not need to write the 3- or 4-lines paragraph. I believe this is not right way to write. The research model is quite interesting, but author(s) need to give more theoretical justifications.

Authors´ answer: The introduction has been restructured and the short paragraphs removed.

Reviewer´s comment:  In the literature review part, there are many problems in this part. Please add more justifications. This section has used recent and relevant cites, but even than arguments are less telling. At certain places, it is emphasized that ‘who’ said ‘what,’ but ‘why’ one has said so is ignored at all. This is not enough for the manuscript. I am not satisfied about the literature review. First of all, author(s) should start the literature from theories, how they can connect with your variables.

Authors´ answer: The introduction has been restructured and rewritten to make clearer the theoretical background of the study

Reviewer´s comment:  Author(s) have used distributive justice, why they did not use further dimensions of the organizational justice? Give the answer?

Authors´ answer: A paragraph has been added in the limitation section describing previous research has focused on other dimension of justice (e.g., interpersonal) whereas the distributive justice, despite the evidence on its core role in job well-being, has been barely analyzed. We totally agree with the author the research will benefit from the joint analysis of the different dimension of attribution. This idea has been included as a suggestion for future research.

Reviewer´s comment:  All hypotheses are confusing please write the simple and proper way.

Authors´ answer: The hypotheses are written in correspondence with the strategy of analysis. Some has been reworded (correlated instead impact) to make more explicit the assertion.

Reviewer´s comment:  The author(s) has tried to anchor the theory at different places, but could not successfully demonstrate the grasp over COR. This has given me an impression that perhaps the relevant literature is not efficiently summarized, analyzed, and examined.

Authors´ answer: Section 1.3 on COR has been reformulated.

Reviewer´s comment:  Results and data collection are not incorrect. Please explain the right way.

Authors´ answer: We have reviewed on the journal to make sure our section follows the same pattern. American Psychological Association rules has been used as a guide for the redaction of the results. More precise indications from the reviewer will be appreciated to improve the paper.

Reviewer´s comment:  Implications: This is very important section for this manuscript. Please add two paragraphs of theoretical and practical implications. I am afraid of this section must be improved, otherwise this cannot be justify your manuscript.

Authors´ answer: The discussion includes now this two paragraphs.

Reviewer 2 Report

The paper is ambitious and in some ways is its only downfall. Prior to publication the authors may benefit from taking time to reviewing the introduction to edit inappropriate paragraph breaks and see if any content could be cut a bit. The introduction itself is incredibly long and does not match that of the discussion. The authors should refer to their discussion and implications and focus on those specific areas in the literature review so the article itself is a bit more focused. 

Author Response

Reviewer´s comment Prior to publication the authors may benefit from taking time to reviewing the introduction to edit inappropriate paragraph breaks and see if any content could be cut a bit.

Authors´ answer: The introduction has been rewritten, restructures and shortened.

Reviewer´s comment: The introduction itself is incredibly long and does not match that of the discussion. The authors should refer to their discussion and implications and focus on those specific areas in the literature review so the article itself is a bit more focused. Prior to publication the authors may benefit from taking time to reviewing the introduction to edit inappropriate paragraph breaks and see if any content could be cut a bit. The introduction itself is incredibly long and does not match that of the discussion.

Authors´ answer: following other reviewer´s suggestion, the first section of the introduction has been shortened. Also, the text has been edited and proofread. The introduction and discussion section has been aligned.

Reviewer´s comment: The authors should refer to their discussion and implications and focus on those specific areas in the literature review so the article itself is a bit more focused.

Authors´ answer: The discussion section has been revised

Reviewer 3 Report

The article presents an interesting and current theme, having simultaneously a good literature review, sample and analysis of results.

Strengths: literature review, methodology, analysis of results and conclusions

Weaknesses: article structure

Improvement proposals 

In the introduction

Introduction and literature review are linked and there should be a separation between introduction and literature review. The introduction must define the objective, methodology, research questions, as well as the parts that make up the article.

Discussion

The relevance of this study (study value) should be mentioned for a better understanding of the topic.

"Despite these limitations, we believe that the present study provides meaningful advances in the understanding of the relationship between surface acting and employee's well-being." Authors must describe a concrete way in which aspects and how.

Author Response

Reviewer´s comment: Introduction and literature review are linked and there should be a separation between introduction and literature review. The introduction must define the objective, methodology, research questions, as well as the parts that make up the article.

Authors´ answer: We have now moved the literature review on the sections following the introduction. The introduction includes a description of the objective, methodology, research questions.

Reviewer´s comment: The relevance of this study (study value) should be mentioned for a better understanding of the topic. "Despite these limitations, we believe that the present study provides meaningful advances in the understanding of the relationship between surface acting and employee's well-being." Authors must describe a concrete way in which aspects and how.

Authors´ answer: The sentence has been moved into the discussion section and now is followed by the description of the theoretical and practical implication from the paper.